# Fabrication of UV-Protective Polyester Fabric with Polysorbate 20 Incorporating Fluorescent Color

**DOI:** 10.3390/polym14204366

**Published:** 2022-10-16

**Authors:** Md. Salauddin Sk, Wasim Akram, Rony Mia, Jian Fang, Shekh Md. Mamun Kabir

**Affiliations:** 1Department of Wet Process Engineering, Bangladesh University of Textiles, Tejgaon, Dhaka 1208, Bangladesh; 2College of Textile and Clothing Engineering, Soochow University, Suzhou 215123, China; 3School of Chemistry and Chemical Engineering, Wuhan Textile University, Wuhan 430200, China

**Keywords:** ultraviolet protection, polyester fabric, polysorbate 20, fluorescent dye

## Abstract

Technological advancement leads researchers to develop multifunctional materials. Considering such trends, this study aimed to conjugate dual functionality in a single material to satisfy aesthetic and functional necessities. We investigated the potentiality of polysorbate 20 to perform as an effective ultraviolet absorber to develop UV-protective fabric. Coumarin derivative (Benzoxazolyl type) disperse dyes are well-known as fluorescent colors. On the other hand, luminescence materials are conspicuous and viable for fashion trends. Deliberate utilization of this inherent property of the dye and incorporation of polysorbate fulfilled the need for dual functionality. In addition, the knitted fabric structure enhanced wearing comfort as well. The effect of polysorbate consolidated the PET fabric as an excellent UV absorber, exhibiting an ultraviolet protection factor (UPF) of 53.71 and a blocking percentage of more than 95% for both UVA and UVB. Surface morphology was studied by scanning electron microscope (SEM). Fourier transform infrared spectroscopy (FTIR) with attenuated mode was used to investigate chemical modification. Moreover, X-ray diffraction (XRD) investigated the crystallography of the surface. Reflectance spectrophotometric analysis unveiled the color strength (K/S) of the dyed polyester fabrics. Finally, light fastness assessment revealed that the developed samples could resist a certain amount of photo fading under a controlled testing environment with the increment of ratings towards betterment.

## 1. Introduction

Sunlight radiation is the combination of ultraviolet, infrared, and visible spectrums. The UV spectrum is significant in all aspects and has the potential to cause detrimental effects on human organs. Ultraviolet radiation (UVR) is subdivided into ultraviolet-A (UVA (315–400 nm)), ultraviolet-B (UVB (280–315 nm)), and ultraviolet-C (UVC (100–280 nm)). Nearly 90–99% of UVR that touches the earth’s surface is UVA, while UVB is only 1–10%. UVR adversely affects the skin, causes cellular damage, and alters immunologic functions. In addition, it results in damaged DNA (by forming cyclobutene pyrimidine dimers), immune suppression, inflammatory responses, and oxidative stress leading to skin cancer and photoaging of the skin [1]. UVA radiation promotes the carcinogenesis of skin stem cells. UVB radiation damages DNA, resulting in tumorigenesis and inflammatory responses. UVR concentration can be increased or decreased depending on various factors, among the prominent ones being the ozone layer, which lies in the stratospheric ozone, providing a thin shield, and protecting us from these rays. Total UVC radiation is absorbed by this protective layer. In contrast, most of the UVB radiation and very little UVA radiation are seized only [2,3]. With the depletion of stratospheric ozone, the number of skin cancer patients is increasing dramatically. Researchers revealed a relationship between ozone layer depletion and the number of skin cancer cases, although the modern lifestyle makes things even worse [4]. Skin is the vital organ of the human body that acts as a barrier, protects from moisture and warmth, and controls body temperature to ensure physical well-being. UV radiation has the potential to penetrate and become absorbed in all three layers of human skin [5]. Apart from avoiding exposure to the sun mainly in peak hours, sunscreens and sunblock creams are frequently used against UVR.

Textiles are designed also for protection against environmental variables such as humidity, temperature, UV radiation, bacteria, etc. Considering these aspects, textile manufacturers focus on designing fabrics that are resistant to environmental undesirable conditions, in addition to providing and enhancing physiological comfort by adopting a low-cost functionalizing process [6]. In recent years, various approaches have been designed to impart new properties to textiles such as electrospinning, electro-deposing, layer-by-layer (LBL), exhaust dyeing, and many more. However, aside from their advantages these processes also have limitations, and must be carried out as per the requirements and demands of practical applications [7,8].

Furthermore, textile materials are appreciated for their flexible structures, low costs, and porous and lightweight characteristics. Chemical modifications and surficial treatments consolidate the capability of textiles to protect against UVR [4,9,10,11,12,13,14]. Various methods such as dyeing, printing, urethane finishing, de-lustering, or UV absorption treatment play a vital role in comfort and protection against the UVR to the skin while worn [5,15]. Figure 1 depicts a general overview of the protection of textile materials owing to their structural characteristics against solar radiation. The accumulative absorption and reflection are causes of such phenomena. Because of the latent structure, both diffused and specular types of reflection take place on the textile’s surface. On the other hand, transmission happens as well. However, because of the influential factors, a certain amount of UV radiation is transmitted and can be absorbed by our skin. A UV absorber substantially reduces the extent of UV contact with the skin and protects against health hazards to the wearer [16]. Research reveals that protection against the UVR of fabric is dependent on fabric color, its weave, finishing method, and additives present, as well as the laundering process [17]. An aromatic polyethylene terephthalate (PET) called polyester has high hydrophobicity and is known to have a high protection factor against UV light transmittance [18]. Polyester fabric has more demand in the current era due to its comfort, active thermal regulation, breathability, light weight, and ease of availability. It is widely used in various sportswear, health/fitness wear, and workwear. Studies reveal that practicing tennis, sailing, scouting, and playing baseball have high UV exposure [19].

Dyes are utilized for the selective absorption of visible light. Most of the dyes absorb light in the region of 400 nm and 700 nm, while some dyes absorb light near the ultraviolet region. In textiles, selected dyestuffs mostly impart a blocking impact against UV light transmittance. For polyester fibers, mostly dispersed dyes are preferred. Various batch-wise dyeing processes are mainly used for woven fabrics [20]. However, exhaust dyeing is the most familiar method for applying disperse dyes to polyester knitted fabric. After the completion of the dyeing cycle, the substrate is rinsed with water to wash off the unfixed dyestuffs [21,22].

Thus, the purpose of this study is the treatment of polyester fabric with a fluorescent dye along with polysorbate in different concentrations to investigate synergistic protection against UV radiation. Polysorbate is a nonionic class of surfactants with some derivatives including polysorbate 20, polysorbate 60, and polysorbate 80. Polysorbate 20 can be synthesized from mineral oils, which makes it relatively nontoxic in nature. Such characteristic makes it feasible to use it as an additive to produce cosmetics, food additives, the pharmaceuticals, sunscreens, etc. On the other hand, polyester fabric is widely used in the textile industry, due to having some unique inherent properties such as better tensile strength and dimensional stability, which are fundamental requirements to be used as sportswear. Protection from the sun is another essential phenomenon for this arena of application. Coumarin dyes are inherently luminescent with UV protection capability as well [23]. A previous study found that mineral oils are viable as UV absorbers [24]. Other studies reveal that natural materials chitosan and zeolite can be used to fabricate functional material for effluent removal [25,26]. Moreover, the natural polymer gum acacia was successful as a capping agent to synthesize nanosilver [27]. Considering all this, polysorbate 20 was selected for this study as a sustainable option. Thus, the purpose of this study is to investigate the potentiality of the deliberate combination of polysorbate 20 and coumarin dye for a synergistic effect as latent UV absorbers. Moreover, the selection of the exhaust dyeing method eliminates the need for additional processes and time.

The exhaust dyeing method was employed considering the defined parameters. Meanwhile, the UPF rating, morphology, and functional groups of the treated and untreated polyester fabric were analyzed as well. Interestingly, it was found that the presence of polysorbate does not affect dyeing quality. Finally, the dyed polyester fabric showed a higher UV protection performance of 53.7 UPF. It was concluded that with the increasing concentration of polysorbate the UV protection of the fabric became better. Furthermore, this treated fabric can be used in various sportswear for protection against carcinogenic UVA and UVB radiation.

## 2. Experimental Details

### 2.1. Materials

Ready for dye (RFD) single jersey polyester (PET) knitted fabric (100%) with a linear density of 160 (g/m^2^) was collected from Fakhruddin Textile Mills Limited, Dhaka, Bangladesh. Commercially available fluorescent disperse dye Rifalon Yellow 10 GN (C.I. Disperse yellow 184-1, Coumarin derivative) was purchased from Rifa Industrial Company Limited, South Korea. Commercial polysorbate 20 (CAS No 9005-64-5) and lab-grade acetic acid were purchased from Rainbow chemical company, Taiwan, and Sigma Aldrich, China, respectively.

### 2.2. Methodology

An exhaust dyeing technique was followed to develop all the samples. Dyeing and polysorbate incorporation were carried out simultaneously [22,28]. For better pipetting precision, 1% stock solutions were prepared in favor of dye, chemical, and acid. The treatment baths were prepared each time for a 5 g fabric sample considering a material-to-liquor ratio of 1:10, and processing was carried out at 130 °C for 60 min maintaining a pH of 4.5. To quantify the degree of UV protection, non-identical concentrations (1%, 2%, and 3%) of polysorbate were considered. Similarly, five unlike dye proportions (1%, 2%, 3%, 4%, 5%) were studied as well. Control samples were also developed to compare the contrast. The amount of different chemical concentrations used for the fabrication of UV-protective polyester fabrics is summarized in Table 1. Once the bath was prepared, a fabric sample was added and the machine started running with a pre-set program accordingly [22,29]. Cold rinsing with de-ionized water was followed by dyeing to remove residual dyes and auxiliaries. Finally, the samples were dried at 100 °C for 20 min with the help of an oven (James heal, Halifax, UK) and characterized for assessment.

### 2.3. Instrumentation

An infrared heating laboratory dyeing machine (SDL, Maidenhead, UK) was used to prepare the UV-protective PET fabric. A digital pipette (Mettler Toledo, Columbus, OH, USA) was used to measure the required proportion of dyes and auxiliaries. A digital pH meter (Mettler Toledo, USA) was used to check the bath alkalinity. The prepared sample was dried using a laboratory oven (James heal, UK).

### 2.4. Measurement and Characterization

The dyed PET fabric (control) and polysorbate 20 incorporated PET fabric’s UPF was determined by using a UV-2000F UV transmittance analyzer, Labsphere U.S.A. Surface analysis was carried out by scanning electron microscope (SEM, JEOL JSM-6460LV, Tokyo, Japan), Fourier transform infrared spectroscope (FTIR, TENSOR 27, Bruker Corporation, Billerica, MA, USA), and X-ray diffraction (XRD, Bruker D8 Advance Diffractometer, Bruker AXS, Karlsruhe, Germany). The characterization processes and evaluations were performed as mentioned in our previous studies [30,31].

## 3. Results and Discussion

### 3.1. Mechanism of the Interaction of Dye Polysorbate 20 Fabric

The possible interaction between dye, polysorbate 20, and fabric samples is shown in Figure 2. Because of the inherent chemical characteristics of the surfactant, polysorbate easily becomes oriented on the surface of the dyebath and forms micelle together with coumarin dye (C_19_H_18_N_2_O_3_) molecules, as investigated in our previous study [22]. The hydroxyl groups (-OH) present in the tail of the polysorbate confer solubility. In such cases, polysorbate acts as a host and coumarin dye as a guest molecule. Another fact is that polysorbate is a long-chain hydrocarbon and has a higher molecular weight compared to the coumarin dyestuff, which is helpful as well for the preceding. Under the acidic condition, the ester group (-R-COO-) of PET fabric will undergo an inclusion complex reaction with polyoxyethylene sorbitan monolaurate molecules (-O-CH_2_-CH_2_-O-) through hydrogen bonds, Van der Waals forces, and electrostatic interaction, and become a part of the host polymer chain [32]. This chemical interaction is caused by π–π* and n–π* transition from the lowest energy to a higher energy level. Various concentrations of dye with polysorbate 20 for protection against UV are listed in Table 1. Control samples were just dyed with fluorescent dye without the addition of polysorbate.

### 3.2. Measuring of UV-Protective Performance of Treated Fabric

UPF analysis of the polyester fabric was carried out with in vitro methodology by utilizing an Ultraviolet Transmittance Analyzer UV-2000F as per the standard EN 13758-1:2002, with the approach method of AATCC 183:2004. This operates by the diffusion of transmittance of the textile sample as a function of the UV spectrum in the region of 250 nm to 450 nm and determines the spectral transmittance, critical wavelength, UVA: UVB ratios, and UPF. All fabric samples were tested at four different locations for covering the entire area of the fabric [33]. After taking a blank scan for the collection of light by integrating the sphere into the optics train lower chamber, calculated as the transmittance of 100% value, a void of sunscreen material, once a 5 cm × 5 cm fabric sample was scanned, non-reflected UV radiation with the combination of sunscreen material was measured. Those radiations were noted by the spectrophotometer and compared with blank scans, where the transmittance of the sample scanned was equivalent to the ratios of beams analyzed from the scanned sample and blank scan. The transmittance spectra were generated as per Equation (1).
(1)$$T\left(\lambda \right)=\left(\frac{S2}{B2}\right)\left(\frac{B1}{S1}\right)$$
where *S*1 and *S*2 are the recordings of the spectrophotometers, and *B*1 and *B*2 are blank scans in the spectrum of 250–450 nm. Once the dark scan data are integrated into the equation, it can be as Equation (2).
(2)$$T\left(\lambda \right)=\left(\frac{S2-SD2}{B2-BD2}\right)\left(\frac{B1-BD1}{S1-SD1}\right)$$

Here, the *BD*2 and *SD*1 were scanned during blank scans and scanning with the samples. The UV analyzer automatically calculated these readings and delivered them in graphical and text format. Here, the mean values of the control and dye-treated sample are depicted in Figure 3 and Table 2. The results reveal a similar trend of UPF rating for control dyed samples of 1%, 2%, 3%, 4%, and 5% as 29, 31, 32, 33.3, and 33.7, respectively. Previous studies revealed that fluorescent dyes have inherent UV protection characteristics [34]. Once the fabric was treated with the nonionic polysorbate-incorporated dye, the UPF readings changed to 43.92, 43.14, 44.48, 46.52, and 53.71, respectively. These results identify that with the addition of polysorbate the UV protection performance of the polyester fabric was improved substantially. Better UV protection performance of polyester fabric may be due to the possible complex interaction of ester groups (-R-COO-) of polyester fibers with the polyoxyethylene sorbitan monolaurate molecule (-O-CH_2_-CH_2_-O-) under acidic conditions, as discussed earlier. Figure 3 depicts the UV-protective performance of various dye concentrations with the addition of polysorbate. The 5% dye incorporated with 3% polysorbate shows the highest UPF rating and best performance, which is 53.71. Usually once the UPF of fabric is above 40 it is UV protective.

Figure 4 explains the mean transmittance rate of polyester fabric. Figure 4A shows where no polysorbate (referred to as control samples) was added, and the higher transmittance rate of UVB is 2.72% with a standard deviation of 0.84 and for UVA is 3.58% and 2.10, respectively. This can be compared to Figure 4B, where polysorbate was 1%, and the cumulative mean transmittance of UVA is 2.48% and UVB is 2.10% with standard deviations of 3.22 and 1.62, respectively. In Figure 4C, when the polysorbate concentration was 2%, the cumulative mean transmittance for UVA was found as 2.30% with a standard deviation of 2.32, and the UVB transmittance rate was 1.98% with a standard deviation of 0.90. Lastly, for the 3% polysorbate concentration, the resulting UVA was 2.28% having a standard deviation of 3.86, while UVB was reported as 1.82% with a standard deviation of 1.22. The results are in agreement with those of the Shindeet et al. report for fluorescent dye [35].

### 3.3. Light Fastness Analysis

A light fastness test is performed to assess the photo-fading resistance of any substrate under standard test conditions (humidity, time, and temperature). The Q-SUN B02 (QLAB, Baltimore, MD, USA) xenon arc chamber was used to investigate the light fastness of the sample according to ISO 105 B02 (2015) [20]. Standard test conditions include irradiance (1.10), relative humidity (40%), black panel temperature (50 °C), chamber air temperature (39 °C), and duration of 24 hrs. Once the test is completed, an assessment is conducted considering the contrast between the unexposed and exposed portion of the samples using the greyscale (James heal, UK). The higher the rating the better the light fastness property of the textile [34,36]. Test results in Table 3 depict that the treated samples have higher ratings compared to the control samples. That is a sign of better light fastness [34]. Test results depict that the incorporation of polysorbate enhances the fabric’s resistance to photo fading compared to those without polysorbate. This means polysorbate molecules are successfully able to resist a certain amount of photo fading.

### 3.4. SEM Analysis

SEM analysis was performed to study the morphological changes at the surface of the PET fabric after polysorbate incorporation. As shown in Figure 5, the untreated samples have smoother surface characteristics compared to the treated ones. This depicts the roughness of morphology with the presence of the dye–polysorbate conjugate at the microscale. The possible reason for the roughness of the polyester fabric may be the interaction of the substrate and the polysorbate molecule. The scratches on the fabric surface were found to be increased after polysorbate treatment, which indicates that the UV protection strength improved with these microscale depositions. This is the outcome of the successful grafting of polysorbate and dye on the polyester fabric surface and performance against UV radiation [37].

### 3.5. FTIR Analysis

The functional group analysis for the surface of PET fabric was performed by Fourier transform infrared spectroscopy (FT-IR) with attenuated total reflectance (ATR) mode and through the accumulating scans of 120 with a resolution of 4 cm^−1^. Figure 6 illustrates the FTIR spectrum of the treated and control PET fabric. The presence of C–H, O–H, C–O, and C–O–C vibrations are characteristic peaks in the range between 1500–800 cm^−1^ [38]. The spectral peaks at 1743 cm^−1^ and 1782 cm^−1^ of control and treated PET fabrics correspond to the stretching vibration of the aromatic ring (-C=O). The peaks at 1433 cm^−1^ and 1478 cm^−1^ were also attributed to the aromatic C-C stretching and C-H in-plane bending, while the peaks at 1324 cm^−1^ and 1352 cm^−1^ were assigned to the CH_2_ wagging. The peaks at 1147 cm^−1^ and 1155 cm^−1^ were ascribed to the C-O vibration mode for the carboxylic ester group present in PET fabric [39]. The peaks at 742 cm^−1^ and 722 cm^−1^ appeared due to the bending vibration of the heterocyclic aromatic ring (=C-H). However, the peak intensity of the treated samples was lower due to the chemical interaction between fabric, dye, and polysorbate 20. A similar phenomenon for the treatment of PET fabric was observed in previous research [36].

### 3.6. XRD Pattern Analysis

The X-ray diffraction patterns of the dyed PET fabric (control) and polysorbate 20 incorporated UV-chemical-treated PET fabric is shown in Figure 7. The findings of diffraction peaks at 2θ of 17.58°, 22.66°, and 25.50° were observed only for the dyed PET fabric. On the other hand, when the PET fabric was treated with polysorbate 20 incorporating fluorescent dyes, the diffraction peaks of 2θ appeared at 17.78°, 22.84°, and 25.50°. Here, all 2θ peaks followed the standard of JCPDS No. 49-2301 for PET fabric [40]. There were no differences between the control and UV-chemical-treated samples. It demonstrated that when original PET fabrics were dyed as well as UV-chemical-treated with polysorbate 20, the major structure of the fabric remained unchanged. A similar phenomenon was also observed for the surface treatment of PET fabric with dilute H_2_SO_4_ [41].

### 3.7. Color Strength Analysis

The color strength (K/S) is an easy way to define the depth of shade. A reflectance spectrophotometer (Datacolor/ 650, USA) was used to investigate the effect of polysorbate concentration on color strength. The instrument was standardized with 2° viewing geometry and a D65 light source with specular components included. Moreover, a larger aperture size (30 mm) was selected to obtain better perfection of reflectance values (R) covering the region of 400–700 nm wavelengths. Finally, K/S values were calculated with the Kubelka–Munk equation (Equation (3)). Table 1 and Figure 7 depict that the K/S values are almost independent of polysorbate concentration with a negligible increment after addition. The K/S values of 1%, 2%, 3%, 4%, and 5% control PET samples were 10.73, 13.26, 13.85, 14.88, and 15.14, which were almost constant after 3% polysorbate incorporation with readings of 11.01, 13.72, 14.83, 15.09, and 15.84, respectively (Figure 8). Therefore, polysorbate does not influence the color strength of dyed PET fabrics.
K/S = (1 − R)^2^/2R(3)

## 4. Conclusions

In conclusion, we have outlined an efficient and easy approach that provides the possibility of incorporating fluorescent dye with polysorbate into the polyester fabric for UV protection properties. Polysorbate was found to perform in a high-temperature dyeing bath, which saves valuable processing time and extra costs for an additional step. The exhaust-dyed polyester fabric demonstrated an excellent UPF rating of 53.7 compared to the untreated rating of 29.04. This will protect the wearer from potential skin cancer effects. The color fastness properties of the dyed fabric were enhanced to 7–8 for 5% dye concentration and 3% polysorbate incorporation. On the other hand, polysorbate concentration had almost no influence on color strength. Moreover, improved UPF of the polyester fabric was subjected to the possible complex chemical interaction of ester groups of polyester with the polyoxyethylene sorbitan monolaurate molecule from the dyeing solution. This approach reveals an easy way to functionalize sportswear which is mostly made of polyester with UV protection properties with the utilization of existing chemistry.

## Figures and Tables

**Figure 1 polymers-14-04366-f001:**
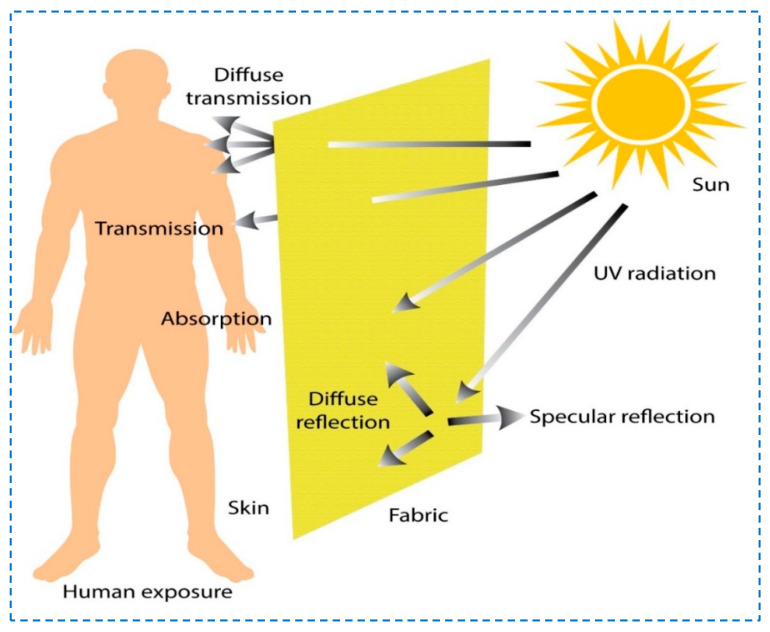
Schematic diagram UV radiations from the sun interacting with textile.

**Figure 2 polymers-14-04366-f002:**
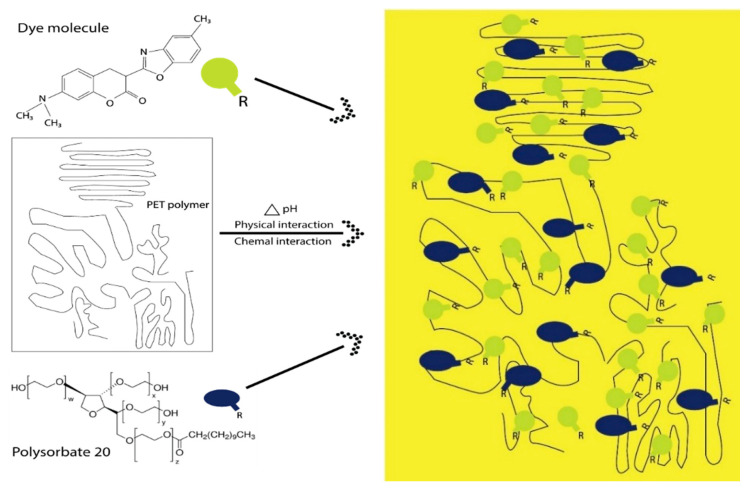
Interaction of dye-mixed polysorbate 20 with polyester fabric during the process of exhaust dyeing.

**Figure 3 polymers-14-04366-f003:**
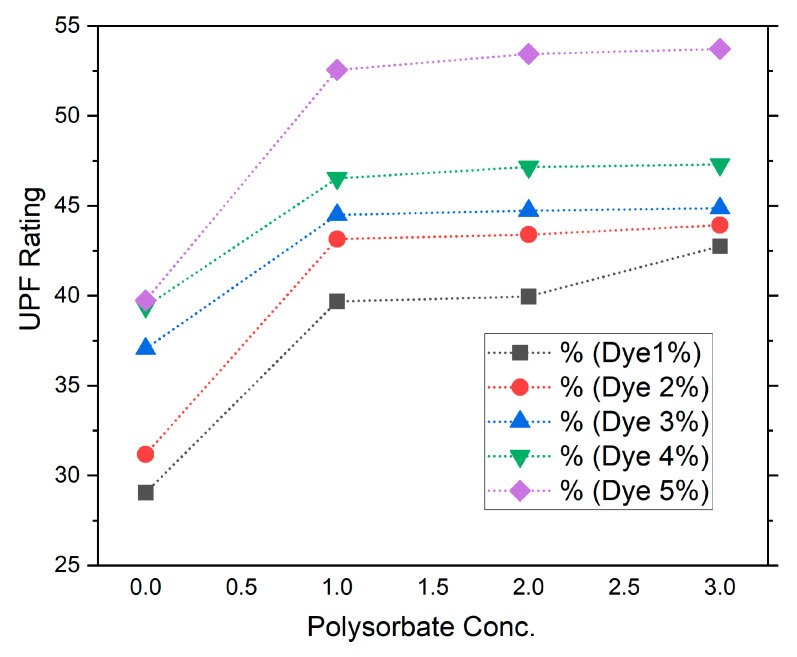
UPF rating of polyester fabric with dye and dye-mixed polysorbate.

**Figure 4 polymers-14-04366-f004:**
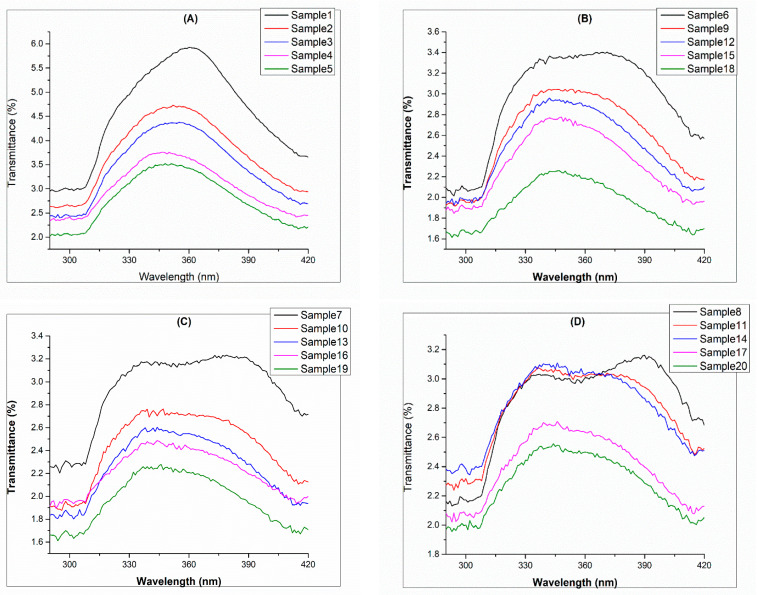
UV protection transmittance of UVA and UVB: (**A**) control samples, (**B**) UV chemical treated with 1%, (**C**) UV chemical treated with 2%, and (**D**) UV chemical treated with 3%.

**Figure 5 polymers-14-04366-f005:**
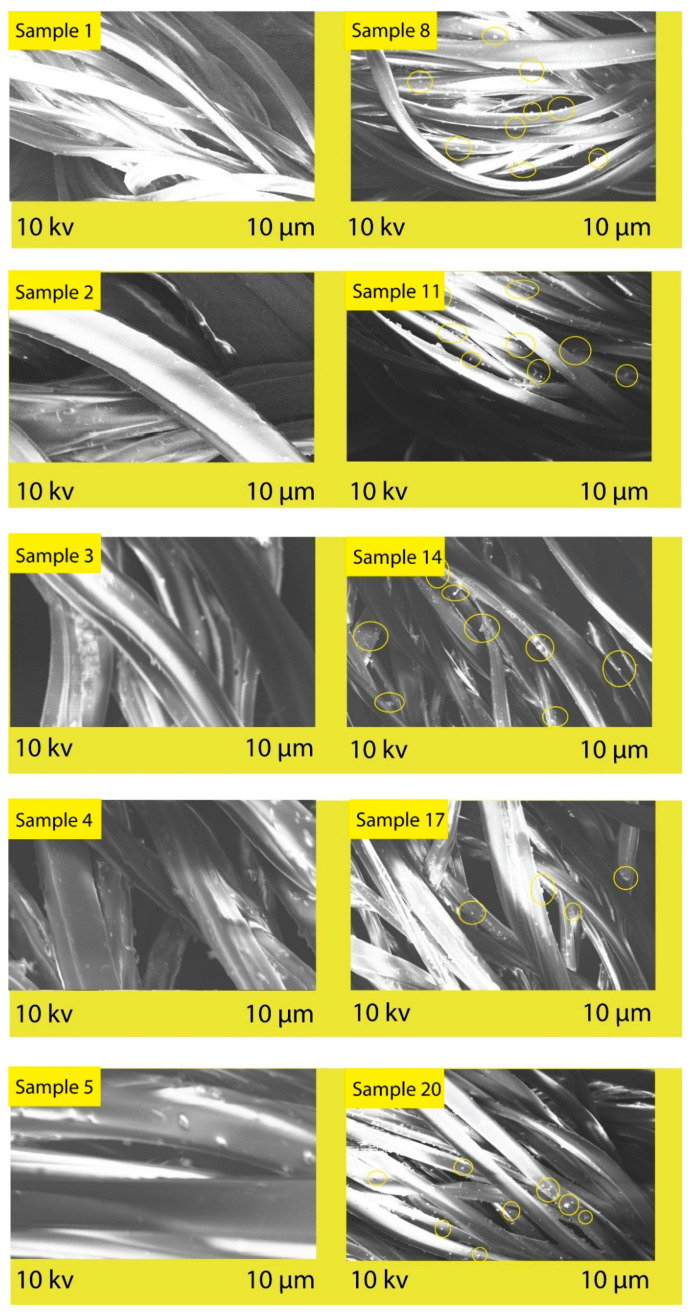
SEM images of control and polysorbate-treated polyester fabric.

**Figure 6 polymers-14-04366-f006:**
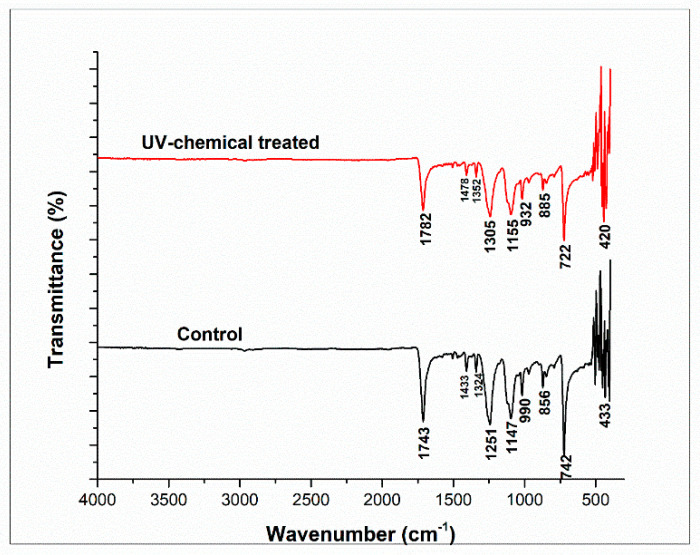
FT-IR spectra of control and dyed polysorbate 20 treated polyester fabrics.

**Figure 7 polymers-14-04366-f007:**
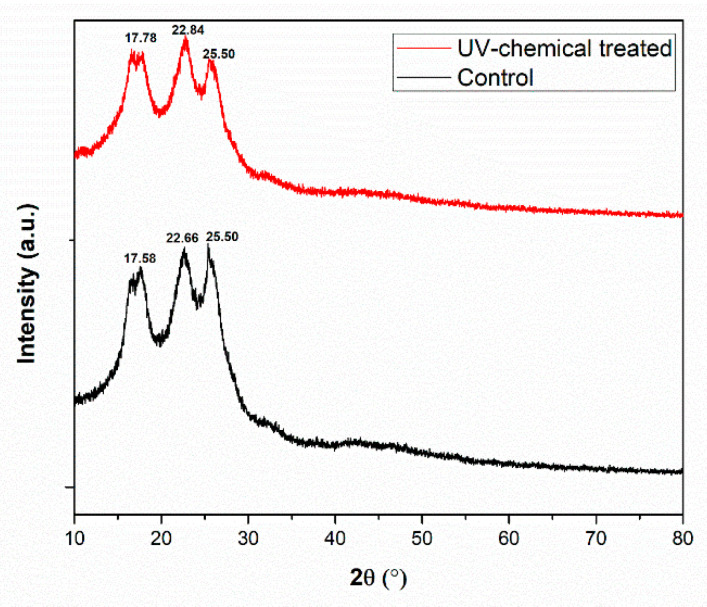
XRD pattern of control and dye polysorbate 20 treated PET fabrics.

**Figure 8 polymers-14-04366-f008:**
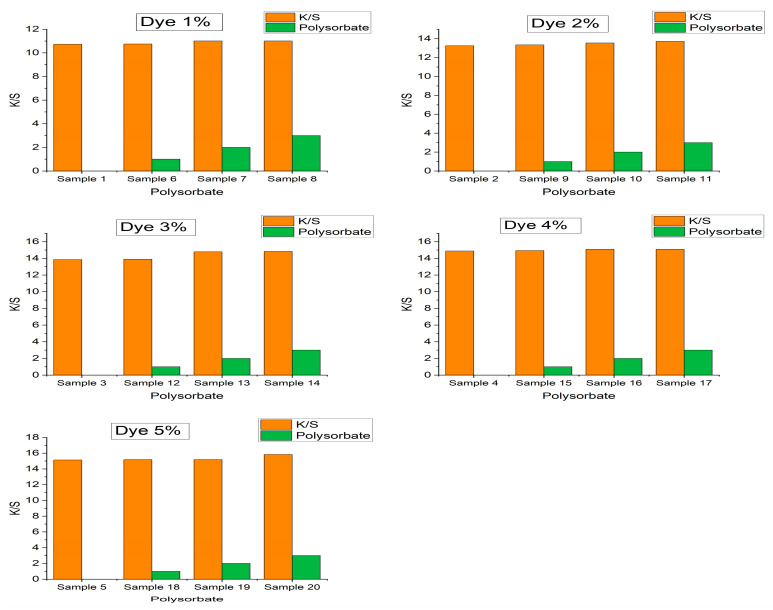
Effect of polysorbate concentration on color strength (K/S) of dyed PET fabrics.

**Table 1 polymers-14-04366-t001:** Various concentrations of dye and polysorbate used for experimentation.

Sample No.	Sample Types	Dye (% owf)	Polysorbate 20 (% owf)	K/S
1	Control	1	0	10.73
2	Control	2	0	13.26
3	Control	3	0	13.85
4	Control	4	0	14.88
5	Control	5	0	15.14
6	Treated	1	1	10.75
7	Treated	1	2	11.01
8	Treated	1	3	11.01
9	Treated	2	1	13.34
10	Treated	2	2	13.55
11	Treated	2	3	13.72
12	Treated	3	1	13.89
13	Treated	3	2	14.78
14	Treated	3	3	14.83
15	Treated	4	1	14.93
16	Treated	4	2	15.09
17	Treated	4	3	15.09
18	Treated	5	1	15.19
19	Treated	5	2	15.19
20	Treated	5	3	15.84

Here, owf means “on the weight of the fabric”.

**Table 2 polymers-14-04366-t002:** UPF ratings of dyed (control) and dyed with the addition of polysorbate 20 (treated) polyester fabric.

Sample No.	Sample Types	UPF Mean	UPF Standard Deviation	UPF Coefficient of Variation
1	Control	29.04	1.48	5.11
2	Control	31.17	3.03	9.73
3	Control	32.05	1.58	4.27
4	Control	33.39	2.19	5.57
5	Control	33.73	3.20	8.07
6	Treated	39.67	4.20	10.59
7	Treated	39.95	0.93	2.34
8	Treated	42.47	5.31	12.50
9	Treated	43.14	2.94	6.82
10	Treated	43.39	5.60	12.90
11	Treated	43.92	1.86	4.25
12	Treated	44.48	5.07	11.40
13	Treated	44.73	4.30	9.63
14	Treated	44.86	4.58	10.21
15	Treated	46.52	2.82	6.06
16	Treated	47.15	1.32	2.80
17	Treated	47.29	7.04	14.88
18	Treated	52.55	5.76	10.97
19	Treated	53.43	4.60	8.62
20	Treated	53.71	5.96	11.09

**Table 3 polymers-14-04366-t003:** Light fastness properties of dyed (control) and polysorbate-dyed (treated) samples.

Sample No.	Sample Types	Light Fastness Rating (1–8) *
1	Control	4
2	Control	4–5
3	Control	5–6
4	Control	6
5	Control	6–7
6	Treated	4–5
7	Treated	5
8	Treated	5
9	Treated	5
10	Treated	5–6
11	Treated	5–6
12	Treated	6
13	Treated	6–7
14	Treated	6–7
15	Treated	6–7
16	Treated	6–7
17	Treated	7
18	Treated	7
19	Treated	7–8
20	Treated	7–8

Rating * (light fastness: 1—poor and 8—excellent).

## Data Availability

The data presented in this study are available on request from the corresponding author.

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
