# Peer review of "Fabrication of UV-Protective Polyester Fabric with Polysorbate 20 Incorporating Fluorescent Color"

_polymers, 2022, doi:10.3390/polym14204366_

Round 1
Reviewer 1 Report
The paper is devoted to the special questions of dyeing of polyster fabrics for sportswear to increase the solar protection. Although the item is actual and important, there several comment to the paper.
- Table 2 contains the excessive amount of meaning signs for UPF mean, UPF standard deviation and variation coefficient.
- All the text needs language and formatting correction. Together with miprints and lost words there are some incorrect wordings in the text: for example, "Vander Wales force" should be changed to "van der Waals force" (line 137).
- Chemical formula of coumarin dye (line 139) is incorrect.
- The characterization processes and evaluation procedures are not described in the paper, only the literature link is available. All the nessesary information should be added to the text.
- The supposed mechanism of bonding of ester functional groups in the fyber structure to the oxiethylene groups is unclear and needs explanation
Taking into concideration all the points, the paper needs the major revision and cannot be published in the present form.
Author Response
Response to Reviewer 1 comments

Reviewer 2 Report
Comments:
0. Major revision. 1. The novelty of this study should be inserted in the text clearly. 2. The advantages and disadvantages of this study should be investigated. 3. The stability of the UV protective polyester fabric should be studied in detail. 4. The mechanism of UV protection should be presented in detail. 5. The “introduction” section of the manuscript can be strengthened and supported with some papers related to the literature and cited (optional for authors): Carbohydrate polymers 237 (2020), 116128; Journal of Molecular Liquids 282 (2019), 115-130; Materials Research Bulletin 47 (2012), 4403-4408; Research on Chemical Intermediates 41 (2015), 3743-3757; Clean–Soil, Air, Water 39 (2011), 673-679; Fibers and Polymers 16 (2015), 1925-1934; Desalination and Water Treatment 57 (2016), 24378-24386; Journal of the Taiwan Institute of Chemical Engineers 81 (2017), 239-246; Journal of Industrial and Engineering Chemistry 32 (2015), 85-98; Colloids and Surfaces A: Physicochemical and Engineering Aspects 355 (2010), 183-186; Journal of applied polymer science 106 (2007), 267-275; Corrosion Science 51 (2009), 1817-1821.
Author Response
Response to Reviewer 2 comments

Reviewer 3 Report
1- Line 90, are reactive dyes suitable for polyester dyeing? Polyester can be dyed only with disperse dyes. Lines 90-96 are correct about cotton, not for polyester.
2- Before line 96, you should explain about polysorbate 20, its structure and why it is able to absorb UV. I found nothing about the application of this surfactant as UV absorber.
3- Line 137, Vander Waals force is correct. There are several spelling and grammatical mistakes in this paper. It is necessary to revise it carefully by an expert.
4- In my opinion, polysorbate 20 has no important effect on UV absorption. It is only a surfactant which may improved the dye sorption of PET. I recommend to study the effect of different concentrations of polysorbate (without dye) on UPF. Also, compare the color strength (K/S) of the samples dyed with or without polysorbate with different concentrations of the dye. If two samples with the same K/S, one with polysorbate and one without it, can be used for comparison and confirming the real effect of this surfactant. All discussions about the effect of this chemical may be changed in this way.
5- FTIR spectra of the samples are very similar. The discussion is not good.
6- SEM images shown nothing important. Images with higher magnification and quality are needed. The discussion about the SEM results is very weak. How the polysorbate can scratch the PET fiber? Why the dyes or chemicals have diffused to the fibers and are present on the surface?
Author Response
Response to Reviewer 3 comments

Round 2
Reviewer 1 Report
The authors have made the significant corrections in the paper, the information on the experimental conditions and on the reaction mechanism was also added. Although there are still some misprints and grammar mistakes in the text (for example, lines 77, 98 and others), this problem may be solved with proofreading. The paper may be accepted for publication after minor corrections.
Author Response
Reviewer comment answer

Reviewer 2 Report
Accept
Author Response
Reviewer comment answer

Reviewer 3 Report
Accept.
Author Response
Reviewer comment answer
